# Role of Haptoglobin as a Marker of Muscular Improvement in Patients with Multiple Sclerosis after Administration of Epigallocatechin Gallate and Increase of Beta-Hydroxybutyrate in the Blood: A Pilot Study

**DOI:** 10.3390/biom11050617

**Published:** 2021-04-21

**Authors:** Jose Enrique de la Rubia Ortí, Jose Luis Platero, María Benlloch, Lorena Franco-Martinez, Asta Tvarijonaviciute, Jesús Escribá-Alepuz, Sandra Sancho-Castillo

**Affiliations:** 1Department of Nursing, Catholic University of Valencia San Vicente Martir, 46001 Valencia, Spain; joseenrique.delarubi@ucv.es (J.E.d.l.R.O.); sandra.sancho@ucv.es (S.S.-C.); 2Doctoral Degree School, Catholic University of Valencia San Vicente Martir, 46001 Valencia, Spain; joseluis.platero@mail.ucv.es; 3Interdisciplinary Laboratory of Clinical Analysis, Campus of Excellence Mare Nostrum, University of Murcia, 30100 Murcia, Spain; lorena.franco2@um.es (L.F.-M.); asta@um.es (A.T.); 4Neurophysiology Department, Sagunto University Hospital, 46520 Valencia, Spain; jesusescriba@hotmail.com; 5Institute of Sleep Medicine, 46021 Valencia, Spain

**Keywords:** epigallocatechin gallate, beta-hydroxybutyrate, multiple sclerosis, haptoglobin, interleukin 6, muscle

## Abstract

Here, we report on the role of haptoglobin (Hp), whose expression depends on the synthesis of interleukin 6 (IL-6), related to the pathogenesis of multiple sclerosis (MS), as a possible marker of muscle improvement achieved after treatment with the polyphenol epigallocatechin gallate (EGCG) and an increase in the ketone body beta-hydroxybutyrate (BHB) in the blood. After 4 months of intervention with 27 MS patients, we observed that Hp does not significantly increase, alongside a significant decrease in IL-6 and a significant increase in muscle percentage. At the same time, Hp synthesis is considerably and positively correlated with IL-6 both before and after treatment; while this correlation occurs significantly reversed with muscle percentage before treatment, no correlation is evident after the intervention. These results seem to indicate that Hp could be a marker of muscle status and could be a diagnosis tool after therapeutic intervention in MS patients.

## 1. Introduction

Multiple sclerosis (MS) is a chronic autoimmune disease of the central nervous system (CNS), which has an unpredictable course of disease and can be presented with different patterns, which can be classified mainly as: “relapsing–remitting” presented by the majority of patients and characterized by exacerbations and remissions, which can become a form of “secondary progressive” MS with progressive disability between attacks; “primary progressive” MS, in which progressive disability develops from the onset; and “relapsing progressive” MS, which is present in very rare cases where the disease gradually worsens and, subsequently, leads to discrete attacks [1]. MS causes muscle breakdown and is characterized by high levels of oxidative stress and inflammation causing axonal demyelination [2,3]. As a result of the inflammatory nature of the disease, several inflammatory markers are associated with the symptoms and pathogenesis, including pro-inflammatory cytokine interleukin 6 (IL-6) [4,5], which is also directly related to muscle damage [6]. In addition, expression of haptoglobin (Hp) is produced in response to IL-6 synthesis [7], which is an acute phase circulating protein produced by hepatocytes and white adipose tissue [8]. Therefore, Hp levels in the blood are increased in prooxidant conditions, such as obesity [9] or skeletal muscle damage [10]. However, its activity is also associated on a central level, in MS patients, with the first line of defense of myelin against oxidative activity, of elevated hemoglobin (Hb) levels released [11] as a consequence of the high fragility of erythrocytes evidenced in these patients, and that alter the blood–brain barrier, damaging the basic protein of myelin [12]. This myelin alteration could justify the positive correlation between free hemoglobin levels in peripheric blood and cerebral atrophy in MS [13].

This effect can also occur at the muscle level, so that Hp binds peripherally to free Hb, carrying it to the liver and limiting its prooxidative action in the muscle [8] (Figure 1). Therefore, an increase in Hp in the blood may be related to a protective activity in muscles by preventing extravasation of free Hb in the skeletal muscle tissue, thus inhibiting O2 deficiency in the skeletal muscle tissue [14]. This prevents oxidative damage and would justify its role as an antioxidant molecule, as has already been demonstrated in previous studies [15,16].

In this sense, the intake of antioxidants can improve the inflammation state. Among these, catechins stand out, which are natural polyphenolic compounds that belong to the flavonoid family, and are found mainly in a wide variety of fruits, vegetables and vegetable drinks, but especially in green tea [17]; 100 mL of green tea (1 g of dried tea leaves brewed for 5 min in 100 mL of hot water) contain on average of 67 ± 11 mg of total catechins, from which stands out the great quantity of epigallocatechin gallate (EGCG) catechin with about 30 mg as the most active molecule in tea [18]. As a consequence of this great anti-inflammatory activity, catechins produce a decrease in cytokines, among which we can highlight IL-6 [19]. Consequently, muscle damage is reduced by reversing the metabolic dysfunction of skeletal muscle, making these antioxidants improve the physical performance in mice [20]. EGCG is known for its efficiency in muscle improvement [21], as well as decreasing hemolysis of previously induced red blood cell membranes [22,23]. All these properties allow EGCG to be considered as a therapeutic alternative for improving muscle performance in MS patients [24].

Furthermore, ketone bodies in the blood also show anti-inflammatory properties, outlining the activity of beta-hydroxybutyrate (BHB), which, by activating the HCA2 receptor coupled to the G protein, causes an anti-inflammatory effect. This would explain its neuroprotective activity against strokes and neurodegenerative diseases [25]. Additionally, there is evidence on the impact of ketosis, obtained through nutritional diets on anthropometric parameters, leading to a decrease in fat [26] and being associated with a greater activity of skeletal muscle as a result of the improved energetic performance in the muscles [27]. This evidence shows that ketogenic diets are used to improve neuromuscular diseases that need to mainly restore skeletal muscle [28]. Coconut oil stands out, due to its high levels of medium-chain fatty acids, among the nutrients with the greatest capacity to produce ketone bodies in the blood [29].

With this in mind, the aim of the study was to determine the role of Hp as a marker for muscle improvement after treatment with 800 mg of EGCG administered daily in two intakes, and increased BHB in the blood after the administration of 60 mL of extra virgin coconut oil, divided equally into two daily intakes.

## 2. Materials and Methods

### 2.1. Study Design

A prospective quasi-experimental pilot study was conducted by means of a clinical trial (ClinicalTrials.gov, NCT03740295).

### 2.2. Population Sample

The main MS associations were contacted to obtain the population sample, informing them of the nature of the study. Thirty-four volunteers with MS decided to take part. The selection criteria were then applied to obtain the final sample. The inclusion criteria included the following: patients over 18 years of age diagnosed with MS in the last 6 months and under treatment with glatiramer acetate and interferon beta. The exclusion criteria included the following: pregnant women; patients with tracheotomy, stoma or short bowl syndrome; patients with dementia; patients evidencing alcohol or drug abuse; patients who had suffered a myocardial infarction, heart failure or had symptoms of angina; patients with kidney disease, liver disease, or hemolytic anemia; patients who regularly do intense physical exercise; and MS patients that were included in other studies with medication or another kind of treatment.

### 2.3. Statistical Analysis

A statistical analysis was conducted with the SPSS v.23 tool (IBM Corporation, Armonk, NY, USA). Once the non-normal distribution of the obtained values was verified in the analyzed variables by means of the Kolmogorov–Smirnov test, the Wilcoxon signed-rank test was used to see the differences before and after the intervention. The categorical data were analyzed with the chi-squared test. A *p*-value below 0.05 was considered significant. The data are provided as a mean ± standard deviation, or the number of patients and percentages. In order to establish the correlations, Spearman’s rank correlation coefficient was established.

### 2.4. Procedure

Instructions were given to not change the prescribed diet. In order to verify if each patient was complying with the treatment, they were called by phone once a week, asking whether they had any doubts or problems with the intervention.

### 2.5. Intervention

Once the selection criteria were applied, a sample of 27 MS patients was obtained. Patients received an isocaloric diet to be followed for 4 months (adapted to the individual characteristics of each patient and divided into 5 meals per day: breakfast, mid-morning snack, lunch, afternoon snack and dinner), enriched with 60 mL of extra virgin coconut oil divided into 2 intakes (30 mL at breakfast and 30 mL at lunch), and with 800 mg of EGCG administered in two 400 mg capsules (two times a day, once in the morning and another in the afternoon).

### 2.6. Measurements

Each participant had a blood test at 11:00 a.m. on an empty stomach before and after intervention. Once the samples had been collected, they were centrifuged to separate the serum, from which the IL-6 concentration was determined using the ELISA technique. BHB levels were measured with a commercial kit (Randox Laboratories, Crumlin, UK), and haptoglobin was determined by a colorimetric method (Tridelta Development Ltd., Kildare, Ireland) in an automatic analyzer (Cobas Mira Plus, ABX Diagnostica, Montpellier, France).

The Matiegka formula [30] was used to calculate muscle weight before and after treatment with which the muscle mass percentage was measured.

### 2.7. Ethical Concerns

The study was conducted in accordance with the Helsinki Declaration (Association, 2013), and with the prior approval of the protocol by the Human Research Committee of the University of Valencia of the Ethics Committee (procedure number H1512345043343). In addition, patients included in the study signed an informed consent form after being notified of the procedures and the nature of the study.

## 3. Results

The final population sample of the study was 27 MS patients, whose sociodemographic characteristics are shown in Table 1.

After intervention, as observed in Figure 2, a significant increase in BHB is produced caused by administering coconut oil in the diet. Despite the increase being significant, a ketosis state was not achieved [31] since the levels increased to an average of 0.10 Mmol/L; however, it should be taken into account that only the concentration of one of the ketone bodies (BHB) was determined, and that it started from very low levels (0.5 Mmol/L), possibly due to the obesity that the population showed (with a BMI of 25.97 (Table 1)), which is associated with low ketone body levels in the blood [32].

The significant increase in BHB could be, in turn, associated with the significant increase obtained in patients’ muscle percentages. For this increase, it should be noted that despite being significant, it just represented a meager 1% compared to that presented by the patients initially; however in our laboratory, like the increment shown in Figure 2 we have been able to observe a loss of approximately −0.59% in the same period of time with MS patients who did not receive any kind of treatment [33]. Therefore, the increase observed in our study may be important. In this sense, there are few studies that indicate an improvement in muscle mass percentage with interventions of the same nature; with other types of therapies, such as intense physical activity, improvements of 1.5% in 3 months have been observed [34]. Therefore, these results could confirm the anabolic action of BHB by increasing the muscle observed after following ketogenic diets [28], which could, in turn, be enhanced by the activity of EGCG as described by Kim AR et al. (2017) who proved how this polyphenol significantly increases the size of muscle fibers, thereby regenerating skeletal muscle in situations of muscle deterioration [35]. In addition, our study saw a considerable decrease in IL-6 already observed in our laboratory in a previous study, which could possibly indicate a reduction in the levels of inflammation both from EGCG and coconut (as a source of BHB in blood). These results could coincide with what has already been observed by other authors when administering EGCG, as this polyphenol negatively regulates its gene expression [36], with the evidenced effect of ketone bodies that decrease the synthesis of cytokines in rats by blocking NMDA receptors [37].

Furthermore, after our intervention, the levels of Hp in the blood did not vary (going from 369.09 ± 97.82 mg/dL to 390 ± 91.15 mg/dL). It should be noted that a sustained hemolysis over time, as a consequence of the progression of the disease, is able to saturate the buffering mechanisms of fr-Hb by Hp, which would induce a decrease in its levels, facilitating even more free Hb entry to the CNS through the damaged BBHE [38], as well as greater muscle destruction. Therefore, the fact that Hp levels, after 4 months of treatment, have not decreased but remain, whereas muscle percentages are increased significantly, could be associated with an improvement in muscular activity, possibly as a consequence of the anabolic effect at a muscular level (both from EGCG [21] and the increase of BHB prompted by the intake of coconut oil [27]), being that our hypothesis is that this improvement could be mediated in turn by the neutralization of free HB. As a consequence, this pattern in the secretion of the protein could be associated with its anti-inflammatory effect, as has already been observed in other diseases, as it promotes Hb–Hp binding, thereby inhibiting Hb-stimulated lipid peroxidation and reducing oxidation [39]. Moreover, this is not only considered anti-inflammatory activity, but also immunomodulatory activity by regulating pro-inflammatory mediators [40]. This fact makes Hp activity acquire special relevance in terms of autoimmune diseases. In particular, in MS animal models, when inhibiting Hp synthesis (EAE Hp knockout), it was observed that the disease was much more serious, which indicates that the levels of Hp in the blood have a prominent role in the progression of autoimmune diseases, and specifically in MS [41], which our results can confirm.

Regarding the correlations of the Hp protein, before intervention, the protein shows a positive correlation with the secretion of IL-6, which demonstrates that in the characteristic conditions of MS of high inflammation and muscle destruction (as evidenced in our study with levels well above the normality of IL-6), the higher the secretion of IL-6, the greater the expression of Hp in the blood (also well above the normal values). These results coincide with those published in animal models of diabetic rats that had high oxidative stress and inflammation as a result of the disease, and where a positive correlation was also registered between both molecules [42]. However, despite improvements in inflammatory levels determined by a significant decrease in IL-6 after treatment, this positive correlation is maintained. This would be explained by the mechanism itself in the expression of Hp, which also precisely depends on cytokine secretion [7] (Table 2).

In spite of this positive correlation between IL-6 and Hp both before and after 4 months of treatment, the correlation between Hp secretion and muscle percentage is reversed before intervention (negative correlation); therefore, no kind of relation is registered after treatment. These results could provide information on the fact that before muscle improvement, the activity of Hp protein would be related to muscle destruction, as indicated by other authors [10]. Nevertheless, and as previously discussed, when improving muscle percentage using Hp to neutralize free Hb by decreasing hemolysis, there is no correlation. This could indicate that free Hp levels in the blood are related to muscle destruction as a consequence of inflammation, but not with oxygenation associated with muscle recovery (Table 2).

After analyzing and debating all our results, EGCG and BHB activities related to the analyzed variables allow us to identify the Hp molecule as a possible biomarker both for inflammatory processes linked to muscular destruction, and muscle improvement after some kind of therapeutic intervention of an anabolic and anti-inflammatory nature. However, it should be pointed out as the main limitation of our study the fact that, since it is a pilot and quasi-experimental exploratory study, a control group was not used; furthermore, the population used is limited. This is why we propose to replicate the study with a larger population, using a control group which would receive a placebo to obtain more definitive conclusions.

## 4. Conclusions

To conclude, the quantification of Hp in blood levels could be a diagnosis marker for MS patients, both regarding muscle damage and muscle regeneration associated with lower inflammation caused after treatment, leading to muscle improvement. However, more in-depth studies are necessary to confirm these conclusions.

## Figures and Tables

**Figure 1 biomolecules-11-00617-f001:**
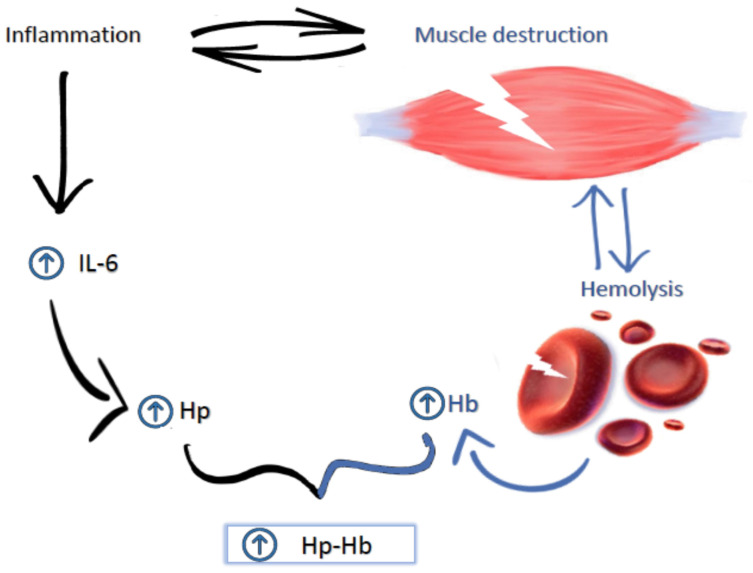
Multiple sclerosis (MS) courses with high inflammation levels characterized by an interleukin 6 (IL-6) increase, on which the expression of haptoglobin (Hp) depends, in turn. In addition to being linked to inflammation, this disease courses with muscular destruction, which is, in turn, related to hemolysis; this hemolysis increases muscular destruction, as a consequence of the free hemoglobin (Hb) pro-oxidative action. On the other hand, a plasmatic Hp increase promotes its union to free Hb (Hp–Hb), decreasing its blood levels and limiting, therefore, its pro-oxidative action.

**Figure 2 biomolecules-11-00617-f002:**
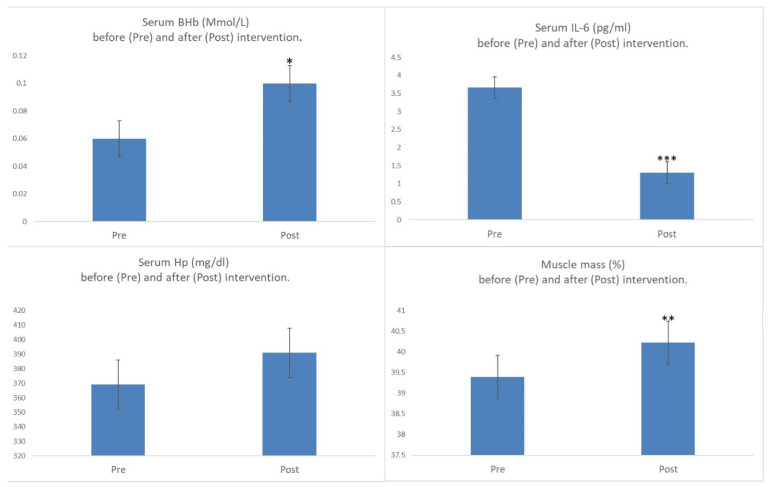
Changes obtained when comparing variables analyzed in serum and muscle percentage of the study population, before (Pre) and after (Post) intervention. BHB: beta-hydroxybutyrate; SD: standard deviation; IL-6: Interleukin 6 (mean value of normal IL-6 1.4 pg/mL); Hp: haptoglobin (HP); *: statistically significant differences (*p* = 0.045); **: statistically significant differences (*p* = 0.03). ***: statistically significant differences (*p* = 0.000);

**Table 1 biomolecules-11-00617-t001:** Sociodemographic characteristics of the study population.

		**Frequency**	**%**
MS Type	Primary progressive	1	3.7%
	Relapsing–remitting	20	74.1%
	Secondary progressive	6	22.2%
Gender	Men	5	18.5%
	Women	22	81.5%
		**Mean**	**SD**
Age (years)		44.56	11.27
Time from MS diagnosis		12	10
BMI		25.97	5.32

MS: Multiple sclerosis; SD: Standard deviation, BMI: body mass index.

**Table 2 biomolecules-11-00617-t002:** Correlations of haptoglobin (Hp) with Interleukin 6 (IL-6) and muscle mass percentage (% muscle), before (Pre) and after (Post) intervention.

Variable	IL-6 Pre		% Muscle Pre	
	Coef	Sig	Coef	Sig
Hp Pre mg/dL	0.657	0.004 **	−0.682	0.001 **
	**IL-6 Post**		**% Muscle Post**	
Hp Post mg/dL	0.527	0.007 **	−0.292	0.148

**: statistically significant differences *p* < 0.005.

## Data Availability

Not Applicable.

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
