# Peer review of "Role of Haptoglobin as a Marker of Muscular Improvement in Patients with Multiple Sclerosis after Administration of Epigallocatechin Gallate and Increase of Beta-Hydroxybutyrate in the Blood: A Pilot Study"

_biomolecules, 2021, doi:10.3390/biom11050617_

Round 1
Reviewer 1 Report
The authors propose that Hp could be a biomarker of muscle status in MS patients. The study is interesting and highlights that Hp could be a diagnosis tool in MS patients after therapeutic intervention. The quality of each experiment is acceptable. However, this reviewer has some comments and concerns about the conclusions reached with the results presented:
- Lines 61 and 65, in the Introduction section, authors should define what catechins and epigallocatechin gallate are.
- Lines 80-81, if this reviewer has not misunderstood, the treatment includes 60 ml of extra virgin coconut oil divided into two intakes. Thus, the authors should add this information in this paragraph.
- Table 1 includes the sociodemographic characteristics of the study population. However, the meaning of “Primary progressive”, “Relapsing‐remitting” and “Secondary progressive” have not been defined before. Can authors add in the text this information?
- Lines 139-140, the authors affirm that the increase of BHB after treatment is caused by the administration of coconut oil. Nevertheless, the patient’s diet was also supplemented with EGCG, which has been demonstrated before that increase BHB. Why authors establish this affirmation? In addition, the difference after treatment is slightly statistically significant, for this reason, authors need more bibliographic support to defend their hypothesis.
- Table 2, since 81.5% of patients were women, could this be the cause of the differences found? Have authors observed gender differences?
- Lines, 157-159 “Therefore, the fact that Hp levels after 4 months of treatment have not decreased but remain, whereas muscle percentages increase significantly, could indicate the efficacy of the intervention at a muscular improvement level, neutralizing free Hb”.
This statement cannot be established with the results obtained. How much muscle mass does a MS patient lose in 4 months? The increase of muscle mass was less than 1%, regardless of whether this result is statistically significant, authors need more data to corroborate their affirmation. In addition, it should be supported by more bibliographic references.
- The authors use correlations of Hp with IL-6 and muscle mass percentage before and after treatment, table 3. In my opinion, because correlation does not imply causality, I strongly recommend including a “control group”, MS patients with a normal diet (not supplemented). If authors want to establish any conclusion or affirmation with their data, they should compare their results of this work with those obtained in control patients with “normal or not supplemented diet”.
Reviewer 2 Report
Introduction and the aim of the study are clearly presented. It should be noted that authors of the work focuses on real aspect. The aim of the study was to assess the role of haptoglobin as a marker for muscle improvement after treatment with EGCG and increased beta-hydroxybutyrate. Assessment statistic methods used does not raise the slightest reservations. Results section include three tables and one figure. It worth pointing out that results interpretation is highly correct.
Row 170 Table 2. Presents very important results, but it would be better to present these results as figure.
The authors cited and presented a lot of available scientific studies. The paper is well written. The presented research may represent the valuable information and I recommend this paper for publication in Biomolecules.
Round 2
Reviewer 1 Report
All comments and suggestions from this reviewer have been considered by the authors I recommend this paper for publication in Biomolecules.